# Impacts of Landscape Type, Viewing Distance, and Permeability on Anxiety, Depression, and Stress

**DOI:** 10.3390/ijerph19169867

**Published:** 2022-08-10

**Authors:** Yun Shu, Chengzhao Wu, Yujia Zhai

**Affiliations:** 1Key Laboratory of Ecology and Energy-Saving Study of Dense Habitat, Department of Landscape Studies, College of Architecture and Urban Planning, Tongji University, Shanghai 200092, China; 2Urban Environments and Human Health Lab, HKUrbanlabs, Faculty of Architecture, The University of Hong Kong, Hong Kong, China; 3College of Architecture and Urban Planning, Big Data and Urban Spatial Analytics LAB, Tongji University, No.1239 Siping Road, Yangpu District, Shanghai 200092, China

**Keywords:** landscape type, viewing distance, edge permeability, depression, anxiety, stress

## Abstract

Contact with nature is beneficial for mental health, including anxiety and stress. Exposure to virtual nature also has similar restorative traits with real nature. However, previous studies on the restorative environment mostly focus on ordinary people while caring less about patients with depressive disorders. Thus, the restorative impacts of virtual nature on patients with depression warrant examination. This research aims to study the restorative effects of virtual reality (VR) landscape type, viewing distance, and permeability on anxiety, depression, and stress in patients with depression. Study A revealed that the perceived restorative level of landscape type varies greatly: grassland > forest > water > undergrowth > urban square. Additionally, natural environments with higher openness, more green elements, more blue sky, and more sunshine exposure had higher restorative levels on perceived depression, anxiety, and stress relief. Study B found that the grassland landscape with a higher viewing distance and a medium vegetation edge permeability provides more restorative impacts for patients with depression.

## 1. Introduction

Increasing evidence demonstrates that the world is suffering from a huge mental health crisis. Mental health conditions contribute to poor health outcomes. These mental disorders are critical precursors to many life-threatening mental and physical illnesses, such as cardiovascular disease, stroke, cancer, type 2 diabetes, and suicide [1,2]. The World Health Organization (WHO) publication “Depression: a global crisis” stated that depression will become the leading health problem worldwide by 2030. Depression and anxiety disorders are the major mental health problems. Over 620 million people suffer from serious depression and anxiety disorders worldwide [3], and cost the global economy USD 1 trillion per year [4]. Furthermore, more than 80% of people experiencing mental health conditions are without any form of quality, affordable mental health care. Little evidence exists on the effectiveness of nature prescriptions, which involve a health provider (e.g., general practitioner) recommending a patient to spend a fixed amount of time a week in a natural setting (e.g., a park) [5].

Contact with nature is beneficial for mental health. In the last decade, a huge number of studies in the fields of environmental psychology, landscape architecture, and urban planning have proven that exposure to nature provides mental health benefits. Attention restoration theory demonstrates that exposure to nature replenishes our attention by capturing our involuntary attention effortlessly [6]. Stress reduction theory argues that contact with nature reduces psychophysical stress [7]. Many studies have confirmed that green open spaces can relieve anxiety and depression, and also arouse positive emotions. Spending time in urban green spaces can increase self-reported friendliness and well-being, and reduce depression or anxiety scores [8]. For patients with depression, contact with nature can mitigate depression, reduce anxiety, and improve cognition [9,10,11,12]. Exposure to natural environments can also reduce subclinical depression and anxiety [13] and improve individual recovery ability [14]. Living near parks and green open spaces is related to lower rates of depression and anxiety [15]. Many studies have also confirmed that natural landscapes can reduce stress and heal the human body. Stress has many negative effects on physical health; under heavy stress, the human immune system is overloaded, antibody production is inhibited, and wound healing will slow down [16]. Therefore, the stress reduction theory can play a significant role in relieving depression.

It has been demonstrated that watching pictures of real nature will improve human health [17], reduce stress, improve mood [18], relieve fatigue [19], and improve cognition [20]. Exposure to virtual nature also has restorative traits similar to exposure to real nature [21]. Watching virtual reality (VR) natural scenery can reduce stress [22,23]. With virtual nature, it is much easier to control disturbing variables than conducting experiments in real nature.

Previous research suggests that exposure to nature is beneficial for stress, anxiety and depression reduction, however, few studies focus on spatial configuration of urban green spaces. We know little about which kind of the virtual nature type, which range of viewing distance, and which range of vegetation edge permeability have the optimal restorative level on stress, anxiety, and depression. Perceived spaciousness was most strongly related by the area over which one could walk [24], and viewing distance is directly related to the space area. Enclosure is important because it influences safety, and perceived enclosure depends on visual permeability [25]. In addition, viewing distance has a huge impact on perceived visual scale [26,27]. The distance between the observer and the landscape will change the scale and quantity of the landscape that the observer can see in the horizontal and vertical space [26], so people’s perception of the landscape will also change accordingly. Li Lin et al. [18] confirmed that preference, pleasure and relaxation are related to lake viewing distance, the vegetation height, mountain height, and building height. Therefore, viewing distance and edge permeability are important perceived spaciousness and enclosure indicators. In addition, participants in previous studies in the fields of landscape and health were mostly ordinary people, children with autism and adults with Alzheimer’s; few were patients with depression.

This research aimed to discover the relationships between urban park types, viewing distance, and edge permeability and anxiety, depression, and stress, and provide evidence-based guidelines for therapeutic landscape design. The research tried to answer the following questions: For patients suffering from depression, does the landscape type have different restorative degree on perceived preference, anxiety, depression, and stress? Which landscape type has the optimal restorative level (Q1)? For patients suffering from depression, do the landscape viewing distance and edge permeability have different restorative degree on perceived preference, safety, anxiety, and depression, as well as physical stress? Which kind of landscape viewing distance and edge permeability have the optimal restorative level (Q2)? What kind of therapeutic landscape design guidelines for patients suffering from depression can we offer based on the research (Q3)?

## 2. Method

### 2.1. Study A: Urban Park Landscape Types and Mental Health

A large number of studies have shown that the natural environment can effectively alleviate anxiety, help healthy people to recover from pressure, and promote cognition and attention. Open space, landscape types with grassland and water bodies have been confirmed to have higher restorative effects. However, few studies have shown that this general rule applies to people with depressive disorders.

This experiment aimed to verify whether grassland landscapes in urban parks with higher openness have better restorative effects on patients with depression.

#### 2.1.1. Participants

Participants were randomly recruited from patients in a third–class, grade A hospital in Shanghai (East Hospital Affiliated with Tongji University). The sample size was 100, and the sample selection rules were as follows:Patient formally diagnosed as having depression by qualified psychiatrists;Depression is not caused by an organic disease;Patient has appropriate cognitive level and is willing to participate in the study and sign the informed consent form.

#### 2.1.2. Stimuli

The theory of “psychological-physiological arousal” regards the visual characteristics of the environment as important factors affecting psychological activities [28,29]. However, there is no unified standard for the classification of urban park landscape types. Boxin Liu divides urban park landscapes into lawn, water body, mountain forest, farmland, and wetland [30]; Meanwhile, Xinxin Wang divides typical urban park landscapes into three categories: lawn, water surface, and avenue [31]. This study combined these classification standards to divide urban park landscapes into six representative categories: grassland, water, forest, undergrowth, urban square, and “other” for uncommon urban landscapes.

Each category was then subdivided based on degree of openness (high, medium, or low). Representative landscape photos are given in Figure 1.

Before the experiment, these photos were printed in color on 200 g copper A5 paper, randomly arranged so the classification scheme would not affect participants, and posted on the wall of the psychological clinic at an average visual height of 1.6 m.

#### 2.1.3. Indicators

In this experiment, a psychological questionnaire and interview were used to qualitatively analyze mood changes and the pressure relief effect resulting from exposure to different types of urban park landscapes. The psychological questionnaire was divided into two parts. The first part focused on the preference of patients with depression and anxiety for landscape types and the effect of pressure relief. The second part concerned factors influencing preference selection, which were divided into three aspects: degree of openness, color, and landscape elements. The psychological interviews used open questions so as to deeply explore the landscape type characteristics that depression patients thought could relieve emotions and pressure.

#### 2.1.4. Procedure and Measures

Samples were collected in the psychological clinics of East Hospital and Tongji University Hospital every day from 20 October 2018 to 3 December 2018, during the working hours of the hospital, specifically from 8:00 a.m. to 11:00 a.m. and 13:00 p.m. to 16:00 p.m.

Each session lasted about 20 min, and the experimental process was divided into three parts, summarized in Table 1. First of all, patients were introduced to the objective and requirements of the experiment (investigation of the impact of landscape type on the health of patients with depression). If the patients were still willing to continue in the experiment, they were asked to sign the informed consent. Second was the evaluation of anxiety and depression degree (mild, medium, or severe) using the self-rating depression scale (SDS) and self-rating anxiety scale (SAS) [32]. Finally, the experiment was conducted. Patients filled in basic personal information, then were guided to consider a series of landscape photos on the wall. Next, they were asked to choose three photos that relieved their depression and relaxed their mind and body greatly. Finally, they were asked to choose the landscape features (openness, color, element) that inspired their choices and also to write down any other contributing factors. At the end of the experiment, patients were thanked for their cooperation.

#### 2.1.5. Results: The Grassland Landscape Has the Optimal Restorative Level

The results indicated that grassland landscapes with high degrees of openness had the greatest positive effect on depression patients. Considering all landscapes, the impact of landscape type on relieving mood and pressure ranked as follows: grassland (31.2%) > forest (27%) > water (23.4%) > undergrowth (12.5%) > urban square (5.8%). Accordingly, an open grassland landscape was considered to be restorative and superior to both open water and semi-open forest landscapes. There was a non-significant difference between the depression degree and perceived restorative preference of landscape types (X-squared = 21.264, df = 5, *p*-value = 0.129).

In terms of openness, patients with depression believed that the landscape with higher openness have higher restoration level on depression, anxiety, and stress. The proportion of people who choose the landscape type with high openness is the largest (56%), followed by medium openness (34%), and the number of people who choose low openness is small (10%). In addition, patients with different levels of depression have different preference for different degrees of open space. Patients with mild and moderate depression prefer high openness much more than medium openness, while patients with no depressive symptoms and severe depressive symptoms believe that high openness and medium openness are equally restorative.

With regard to landscape color, more than half of participants thought that green landscapes made them relaxed and comfortable; the next most common choices were blue, purple, and yellow in that order. Overall, cold colors made it easier for patients to feel relaxed and emotionally relieved. No difference was observed in landscape color preference for patients with different degrees of depression or anxiety.

In terms of preference for landscape elements, lawn and water surfaces were most commonly considered to have restorative effects, followed by trees and tree arrays; many participants also mentioned sky and sunshine. No significant differences in landscape element preferences were observed for different levels of depression and anxiety, which was consistent with the overall preference degree.

### 2.2. Study B: Relationship between Viewing Distance, Edge Permeability of Lawn Landscapes and Mental Health

Study B was based on study A. Study A had already revealed that the grassland type of urban park had the optimal restorative effect on patients with depressive disorders. In addition, it is also demonstrated that higher openness had higher restorative effects. Furthermore, one of the landscape factors that most impacts patient mood and stress relief is spatial openness. In the past decades, researchers have explored edge permeability and the horizontal area of a space as the factors affecting spatial openness [24,25,33]. The horizontal area of a space can be transferred into viewing distance if the space is square. Therefore, the objective of study B was to examine what kind of grassland configuration (viewing distance and edge permeability) had the optimal restorative effect on patients with depressive disorders.

#### 2.2.1. Participants

A statistical power calculation with the assistance of G*Power 3.1.9.4 (Erdfelder, Faul, & Buchner, Germany) was conducted in order to determine the number of participants [34,35,36]. We use both one-way and two-way ANOVA analysis for calculation. The G * power results showed that a sample size of 40 would produce a power value of 0.95. In order to obtain more precise experimental results, we recruited far more participants than calculated. Ninety-nine participants were recruited to participate in this study. The selection criteria were as follows:(1)Diagnosed as having a depressive episode by an attending physician of the third-class grade A hospital or as a potential patient with at least mild anxiety/depression symptoms according to the self-rating anxiety/depression scale (SAS/SDS);(2)Did not use tobacco, alcohol, caffeine, dairy products, etc. for 24 h before the experiment, and did not participate in vigorous exercise for 6 h before the experiment;(3)Willing to cooperate with the experiment and provide informed consent.

#### 2.2.2. Stimuli

This study used a 3 × 3 factorial design with rendered images. Firstly, grassland landscapes were established with three viewing distance scales, namely 20 m, 100 m, and 200 m. These consisted of three square-shaped lawns of 40 × 40 m, 200 × 200 m, and 400 × 400 m, respectively. Viewing points were set in the center of each at a height of around 1.6 m, and the viewing distance remained unchanged when looking around.

Vegetation consisted of common tree species with height and crown breadth of 7 m and 10 m, respectively. The edge permeability of trees was calculated as the percentage of the length of the crown projecting to the ground, and was set at one of three values, namely 30%, 70% and 100%. Tree arrangements were consistent and surrounded the lawn on all four sides.

Using all possible combinations of the selected viewing distance and edge permeability values gave nine landscape types in a 3 × 3 grid. The plans are shown in Figure 2, and the renderings in Figure 3. The space within the edge was covered with grassland, and the space outside the edge by cement pavement. In order to control variables, all elements aside from viewing distance and edge permeability were kept consistent, such as the sky, grass type, tree species, etc. Finally, an urban street without any trees and grassland was added as a control group.

First of all, the pilot study was conducted to test 12 patients. In order to improve the experimental design, the results were used to analyze whether landscapes with different viewing distances and edge permeability had different effects on pressure relief.

#### 2.2.3. Indicators

Physiological indicators such as skin conductance (SC), skin temperature (ST), heart rate (HR/BVP), blood volume amplitude (BVP Amp), blood oxygen value (SPO2), and heart rate variability (RMSSD, SDNN, LF/HF) were used to evaluate the pressure relief effect of experiencing specific landscape scenes on patients.

Psychological indicators, namely the degree of preference, sense of safety, sense of anxiety, and sense of depression, were used to evaluate patient preference for different grassland landscapes and image capacity for depression/anxiety restoration. Each patient scored the landscape scenes on a scale of 0–10 points. For degree of preference and sense of safety, which are positive indicators, a higher score corresponded with greater liking and security. For anxiety and depression, which are negative indicators, a higher score denoted feeling more anxious and depressed.

#### 2.2.4. Equipment

The VR device used was the Oculus Rift (Facebook Technologies, LLC, Menlo Park, CA, USA), which is a head-worn electronic device designed for interactive games. The Rift is equipped with two ocular lenses, each with a resolution of 640 × 800, giving a total resolution of 1280 × 800 dpi when the vision of both eyes is combined. The Rift creates an immersive stereoscopic image through splitting the screen into two parts, with the left-eye image displayed on the left side and the right-eye image displayed on the right side.

Physiological indicators were measured with a NeXus-10 biofeedback instrument (Mind Media, Herten, Germany), which is a multifunctional eight-channel system integrating biofeedback and neural feedback. It is capable of measuring and giving feedback for multiple physiological signals simultaneously.

#### 2.2.5. Procedure and Measurements

As this experiment required a significant amount of equipment and had high demands in terms of environmental factors, it was necessary to conduct the experiment in a relatively quiet environment to eliminate interfering factors and ensure it was carried out uniformly. Therefore, the experiment was conducted in a studio; participants made appointments and came to the studio to take part in the experiment at the appointed time.

Preparation: First of all, the patients were asked to fill in the background information form and sign the consent form. Then, the experimental objective and procedure were introduced to participants. After that, the patients were asked to sit in front of the computer and wear the VR helmet, then to keep sitting calmly for 3 min with their eyes closed, until physiological values such as skin electricity and skin temperature fluctuated normally. The biofeedback equipment was connected through the whole experiment (see Figure 4).

Pressure test stage: A simple pressure test--trier social stress test (TSST) [37,38] was adopted to induce moderate pressure, namely by making the participants count in their heads for 1 min. The mental arithmetic method adopts a task of continuous subtraction, reducing four-digit numbers by two-digit numbers, such as 8079-19. The participant is required to state the numbers in sequence, e.g., 8060, 8041, 8022…. When a wrong calculation is made, the process should be restarted in order to increase the pressure on the participant. That this mental calculation brings some pressure upon patients was confirmed in the pilot experiment.

Relaxation stage: In this stage, participants experienced the landscape scene for 1 min. While experiencing the scene, the participants were to completely relax and imagine themselves to be in the landscape environment for real. Although most physiological indicators can be conducted and recorded in a few seconds, the heart rate variability analysis must be recorded for at least 1 min in order to produce an analysis report, hence this stage lasting 1 min. As the physiological indicators are very sensitive, the participants could not speak during the experiment, but they could look around in order to eliminate other interference factors.

After the first pressure increase, participants scored their stress and depression (0–10 points). At the end of the relaxation stage, the participants scored the landscape scene they experienced in terms of preference, safety, anxiety, and depression (0–10 points) according to their current psychological perception. Then, the same procedure was repeated for the remaining eight landscape scenes.

After the experiment, the participants signed and received their reward. The experiment lasted for 30–40 min in total, and the experimental procedure is summarized in Table 2.

## 3. Results

### 3.1. Study A: The Grassland Landscape Has the Optimal Restorative Level

The results have been presented in Section 2.1.5.

### 3.2. Study B: Grassland Landscape with a Higher Viewing Distance and Medium Edge Permeability Has Higher Restorative Level

#### 3.2.1. Overall Impacts of Viewing Distance and Edge Permeability on Psychological Indicators

The analysis of physiological indicators is divided into three steps. The first step was to do data processing: remove discrete values and error values, and test whether the data met the normal distribution through descriptive statistical analysis. In the second step, two-way ANOVA analysis of variance was used to analyze the response of the subjects to 8 psychological indicators—whether skin conductance, skin temperature, heart rate, blood volume amplitude, blood oxygen, heart rate variability SDNN, RMSSD, and LF/HF had a significant impact or interactive impact, and whether the degree of depression had an effect on the results. The third step was to do correlation analysis. The correlation function model was constructed for the variables with significant correlation through curve fitting, and the fitting equation was used to obtain the equation that best matched the data.

#### 3.2.2. The Impacts of Viewing Distance and Edge Permeability on Physiological Indicators

The analysis of physiological indicators was divided into three steps. The first step was to perform data processing, remove discrete values and error values, and test whether the data met the normal distribution through descriptive statistical analysis. In the second step, two-way ANOVA analysis of variance was used to analyze the participants’ physical responses after immersed in landscape settings with different viewing distances (20 m, 100 m, 200 m) and edge permeability (30%, 70%, 100%). The third step was correlation analysis. The correlation function model was constructed for the variables with significant correlations through curve fitting.

According to the two-way ANOVA analysis, there was no significant correlation between the viewing distance and the △ESC, F(2,889) = 0.079, *p* = 0.924, η^2^_p_ ^d^ = 0.000 (Table 3). There was also no significant correlation between edge permeability and △ESC, F(2,889) = 0.118, p = 0.888, η^2^_p_ ^d^ = 0.000 (see Table 3). There was also no significant correlation between the cross-effects of viewing distance, edge permeability and ΔESC, F(4,889) = 0.865, *p* = 0.484, η^2^_p_ ^d^ = 0.004 (Table 3). η2p d is the effect size, and its value ranges from 0 to 1. The larger the value of η^2^_p_ ^d^, the more the variance effect of the dependent variable is explained. The analysis of other physiological indicators was the same, and will not be described in detail below.

As shown in Table 4, there was no significant correlation between viewing distance and △E_ST_, F(2,889) = 1.538, *p* = 0.215, η^2^_p_ ^d^ = 0.003; there was no significant correlation between edge permeability and △E_ST_, F(2,889)= 0.983, *p* = 0.375, η^2^_p_ ^d^ = 0.002; and the cross-effect of viewing distance × edge permeability also had no significant correlation with △E_ST_, F(4,889) = 0.260, *p* = 0.903, η^2^_p_ ^d^ = 0.001.

As shown in Table 5, from the results of the inter-subject effect test, it can be seen that there was no significant correlation between viewing distance and ΔE_HR_, F(2,889) = 0.131, *p* = 0.877, η^2^_p_ ^d^ = 0.000; there was also no significant correlation between edge permeability and ΔE_HR_, F(2,889) = 0.686, *p* = 0.504, η^2^_p_ ^d^ = 0.002; and there was also no significant correlation between the cross-effect of viewing distance × edge permeability, and ΔE_HR_, F(4,889) = 0.571, *p* = *0*.684, η^2^_p_ ^d^ = 0.003.

As shown in Table 6, from the results of the inter-subject effect test, it can be seen that there was no significant correlation between viewing distance and ΔE_BVP Amp_, F(2,889) = 0.912, *p* = *0*.971, η^2^_p_ ^d^ = 0.002; there was also no significant correlation between sparse density and ΔE_BVP Amp_, F(2,889)= 0.030, *p* = *0*.402, η^2^_p_
^d^ =.000; and there was also no significant correlation between the cross-effect of viewing distance × edge permeability, and ΔE_BVP Amp_, F(4,889)= 0.588, *p* = 0.671, η^2^_p_ ^d^ = 0.003.

As shown in Table 7, from the results of the inter-subject effect test, it can be seen that there was no significant correlation between viewing distance and ΔE_SPO2_, F(2,889)= 0.099, *p* = 0.906, η^2^_p_ ^d^ = 0.000; there was also no significant correlation between sparse density and ΔE_SPO2_, F(2,889)= 0.304, *p* = 0.738, η^2^_p_ ^d^ = 0.001; and there was also no significant correlation between the cross-effect of viewing distance * edge permeability, and ΔE_SPO2_, F(4,889) = 0.339, *p* = 0.852, η^2^_p_ ^d^ = 0.002.

As shown in Table 8, from the results of the inter-subject effect test, it can be seen that there was no significant correlation between viewing distance and ΔE_RMSSD_, F(2,889) = 0.770, *p* = 0.463, η^2^_p_ ^d^ = 0.002; there was also no significant correlation between sparse density and ΔE_RMSSD_, F(2,889) = 1.673, *p* = *0*.188, η^2^_p_ ^d^ = 0.004; and there was also no significant correlation between the cross-effect of viewing distance * edge permeability, and ΔE_RMSSD_, F(4,889)= 0.819, *p* = 0.513, η^2^_p_ ^d^ = 0.004.

As shown in Table 9, from the results of the inter-subject effect test, it can be seen that there was no significant correlation between viewing distance and ΔE_SDNN_, F(2,889) = 1.398, *p* = 0.248, η^2^_p_ ^d^ = 0.003; there was also no significant correlation between sparse density and ΔE_SDNN_, F(2,889)= 0.891, *p* = *0*.411, η^2^_p_ ^d^ = 0.002; and there was also no significant correlation between the cross-effect of viewing distance * edge permeability, and ΔE_SDNN_, F(4,889) = 1.439, *p* = 0.219, η^2^_p_ ^d^ = 0.007.

As shown in Table 10, from the results of the inter-subject effect test, it can be seen that there was no significant correlation between viewing distance and ΔE_LF/HF_, F(2,889) = 0.022, *p* = 0.978, η^2^_p_ ^d^ = 0.000; and there was also no significant correlation between sparse density and ΔE_LF/HF_, F(2,889)= 0.089, *p* = *0*.915, η^2^_p_ ^d^ = 0.000, although the interaction effect of viewing distance * edge permeability was significant with ΔE_LF/HF_, F(4,889)= 2.431, *p* = 0.046 *, η^2^_p_ ^d^ = 0.011. However, since the single factors of viewing distance and edge permeability had no effect on △E_LF/HF_, their interactive effects made no sense.

For physiological indicators, viewing distance and edge permeability had no significant effect (Table 11).

#### 3.2.3. The Impacts of Viewing Distance and Edge Permeability on Psychological Indicators

The psychological index analysis was also divided into 3 steps: the first step was to correct the data and conduct descriptive statistical analysis. The second step was to use two-way ANOVA analysis of variance to examine the participants’ psychological responses after immersed in landscape settings with different viewing distances (20m, 100m, 200m) and edge permeability (30%, 70%, 100%). The third step was to do correlation analysis: to examine how viewing distance, edge permeability, and viewing distance × edge permeability affected the four psychological indicators. Then use the curve fitting to establish the optimal model. Finally, we tested whether the degree of depression had an impact on the results.

For psychological indicators, different viewing distances and edge permeability had no significant impact on sense of safety, but had a significant impact on preference, anxiety, and depression. The impacts of edge permeability on perceived anxiety reduction were more significant, and the impacts of viewing distance × edge permeability on depression were also very significant (see Table 11).

Preference

Viewing distance has a significant effect on preference

Through two-way ANOVA analysis of variance, the effect of viewing distance on perceived preference was significant, F(2,882) = 3.734, *p* < 0.05, η^2^_p_ = 0.009; while the effect of edge permeability on the perceived preference was not significant, F(2,882)= 1.906, *p* > 0.05, η^2^_p_ = 0.004. The interaction effect of viewing distance × edge permeability had no significant effect on preference degree, F(4,882) = 0.320, *p* > 0.05, η^2^_p_ = 0.001, as shown in Table 12.

As shown in Table 13, participants preferred the landscape setting with a viewing distance of 100 m (M = 6.27 ± 1.805), followed by the landscape setting with a viewing distance of 200 m (M = 6.19 ± 1.911). The landscape setting with a viewing distance of 20 m (M = 5.88 ± 1.862) was the least preferred.

As shown in Figure 5, there was a significant difference in the preference degree between the viewing distance of 20 m and the viewing distances of 100 m and 200 m. The relationship between the preference degree and the viewing distance was: 100 m > 20 m, 200 m > 20 m.

The relationship between preference degree (Y) and viewing distance (X) is: “Y = −0.000032X^2^ + 0.009 X + 5.715”. The highest preference degree was achieved for a viewing distance of 141 m. When the viewing distance was between 0 and 141 m, preference degree would increase as the viewing distance increased. When the viewing distance exceeded 141 m, preference degree would decrease as the distance increased (Figure 6).

Anxiety

Viewing distance has a significant effect on anxiety

As shown in Table 14, viewing distances had a significant effect on perceived anxiety, F(2,882) = 4.266, *p* < 0.05, η^2^_p_ = 0.010. Edge permeability had a higher significant effect on perceived anxiety, F(2,882) = 6.237, *p* < 0.01, η^2^_p_ = 0.014; however, the interaction effect of viewing distance × edge permeability was not significant, F(4,882) = 0.470, *p* > 0.05, η^2^_p_ = 0.002. Therefore, perceived anxiety in landscape settings with different viewing distances was not affected by edge permeability, and the perceived anxiety in landscape settings with different edge permeability levels was also not affected by viewing distance.

There was a significant difference in the perception of anxiety between participants with visual distances of 20 m and 100 m (*p* < 0.05); there was also a significant difference between landscape settings of 20 m and 200 m (*p* < 0.05), but there was no significant difference between the 100 m and 200 m groups (*p* > 0.05). The perceived anxiety of the landscape setting with a viewing distance of 20 m (M = 3.18, SD = 1.711) was 0.333 higher than that of the landscape setting with a viewing distance of 100 m (M = 2.85, SD = 1.619), which was 0.350 higher than that of the landscape with a viewing distance of 200 m (M = 2.83, SD = 1.635) (Table 15).

As shown in Figure 7, the participants’ perception of anxiety in landscape settings with different viewing distances, from low to high, was 100 m < 20 m, 200 m < 20 m. It can be inferred that the farther the viewing distance, the lower the perceived anxiety. However, when the distance reached a certain limit, the anxiety levels did not change significantly.

The relationship between perceived anxiety (Y) and viewing distance (X) is: “Y = 0.000022X^2^ −0.007X + 3.310”. Perceived anxiety was lowest when the viewing distance was 159 m. When the viewing distance range is between 0–159 m, perceived anxiety would be reduced as the viewing distance increased. When the viewing distance exceeded 159 m, perceived anxiety would increase slightly as the viewing distance increased (Figure 6).

b.Edge permeability has a significant effect on anxiety

There was a significant difference between the participants’ perception of anxiety in landscape settings with edge permeability of 30% and 100% (*p* < 0.05), and there was also a significant difference between the 70% and 100% groups (*p* < 0.01). There was no significant difference in perceived anxiety between the 30% and 70% groups (*p* > 0.05). For the landscape setting with 30% edge permeability (M = 2.90, SD = 1.667), the perceived anxiety was 0.316 lower than that with 100% edge permeability (M = 3.22, SD = 1.667); for the landscape setting with 70% edge permeability (M = 2.75, SD = 1.936), the perceived anxiety was 0.468 lower than that of the landscape setting with 100% edge permeability (Table 16).

The participants’ perceived anxiety in landscape settings with different edge permeability levels from low to high was: 30% < 100%, 70% < 100%. It can be seen that a space that was too dense had a poor effect on anxiety reduction. However, when it was sparse to a certain extent, the difference in perceived anxiety was not obvious (Figure 8).

The relationship between perceived anxiety (Y) and edge permeability (X) is: “Y = 2.77X^2^ − 3.149X + 3.594 (0 ≤ X ≤ 1)”. Perceived anxiety was lowest when the edge permeability was 57%. When the edge permeability range was between 0–57%, perceived anxiety would be reduced as the edge permeability increased. When the edge permeability range was between 57%-100%, perceived anxiety would increase slightly as the edge permeability increased (Figure 6).

Depression

The effect of different viewing distances on perceived depression was very significant, F(2,882) = 12.498, *p* < 0.001, η^2^_p_ = 0.028; the effect of different edge permeability levels on perceived depression was also very significant, F(2,882) = 9.995, *p* < 0.001, η^2^_p_ = 0.022; however, the interaction effect of viewing distance * edge permeability was not significant, F(4,882) = 0.392, *p* > 0.05, η^2^_p_ = 0.002. Perceived depression in landscape settings with different viewing distances was not affected by edge permeability, and perceived depression in landscape settings with different edge permeability levels was also not affected by viewing distance (Table 17).

Viewing distance has a significant effect on depression

There was a significant difference between the participants’ perception of depression in landscape settings with viewing distances of 20 m and 100 m (*p* < 0.001), and there was also a significant difference in the perception of depression in landscape settings with viewing distances of 20 m and 200 m. (*p* < 0.001), while there was no significant difference in the perception of depression between the viewing distances of 100 m and 200 m (*p* > 0.05). Perceived depression by the landscape setting with a viewing distance of 20 m (M = 3.10, SD = 1.961) was 0.603 higher than that of the landscape setting with a viewing distance of 100 m (M = 2.50, SD = 1.732), which was 0.677 higher than that with a viewing distance of 200 m (M = 2.43, SD = 1.775) (Table 18).

The participants’ perceived depression in landscape settings with different viewing distances from low to high was: 100 m < 20 m, 200 m < 20 m. It can be seen that the farther the viewing distance was, the lower the perceived depression, but to a certain extent, the perceived depression did not change significantly, as shown in Figure 9.

A quadratic function relationship between perceived depression (Y) and viewing distance (X): “Y = 0.000038X^2^ −0.012X + 3.331”. Perceived depression was lowest when the viewing distance was 158 m. When the viewing distance range was between 0–158 m, perceived depression would be reduced as the viewing distance increased. When the viewing distance was further than 158 m, perceived depression would increase slightly as the viewing distance increased (Figure 6).

b.Edge permeability has a significant effect on depression

There was a significant difference between the participants’ perceived depression in landscape settings with edge permeability of 30% and 100% (*p* < 0.001), and there was also a significant difference in perceived depression in landscape settings with edge permeability of 70% and 100%. There was no significant difference between landscape settings with edge permeability of 30% and 70% (*p* > 0.05). For the landscape setting with edge permeability of 30% (M = 2.51, SD = 1.869), perceived depression was 0.556 lower than that of a landscape setting with edge permeability of 100% (M = 3.06, SD = 1.837). For the landscape setting with edge permeability of 70% (M = 2.47, SD = 1.784), perceived depression was 0.593 lower than that with edge permeability of 100% (Table 19).

The participants’ perceived depression in landscape settings with different edge permeability levels from low to high was: 30% < 100%, 70% < 100%. It can be seen that the high edge permeability of the grassland landscape had a lower restorative effect on perceived depression. However, if edge permeability was too low, it would lead to an insufficient green dose and on conversely have a poor effect on perceived depression reduction, as shown in Figure 10.

A quadratic relationship existed between perceived depression (Y) and edge permeability (X): “Y = 2.954X^2^ − 3.047X + 3.153 (0 ≤ X ≤ 1)”. Depression was lowest when the edge permeability level was 52%. When the edge permeability range was between 0–52%, perceived depression would be reduced as the edge permeability increased. When the edge permeability range was between 52%-100%, perceived depression would increase as the edge permeability increased (Figure 6).

## 4. Discussion

### 4.1. Healthy Landscape Open Space Design Guidelines

Paraskevopoulou found that there is still a lack of research on guidelines for the design of healing gardens for patients [39]. There are mainly three categories in terms of healing garden guidelines: one is for patients with general mental illness, another is for Alzheimer’s disease, and the other is for patients with neuropsychiatric diseases. There are currently no specific guidelines for the design of rehabilitation landscapes for people with depression. This study offers guidelines for healthy landscape design for patients with depression. Health-oriented landscape design needs to consider not only general principles, but some specific health indicators, such as landscape type, openness, elements, color, etc. (see Table 20).

### 4.2. Contributions and Implications

Although this study was exploratory, it offers certain referential significance to supplement and improve the existing guidelines for rehabilitation landscape design; the results can provide scientific suggestions for traditional landscape design for healing and supplement health-oriented guidelines for landscape design. One of the innovations of this study was to select patients with depression as the research object, which was an interdisciplinary attempt to combine landscape design with medicine. In most other evidence-based research studies on health-oriented landscapes, the research objects were most often healthy people instead of patients. Similarly, studies on patients mainly focus on the medical field; few enter into the landscape design field. Even when such research focuses on landscape, it is aimed at children with autism or elders with dementia.

Another innovation of this study was the research methodology. The traditional method of experiencing a landscape is to look at photos, watch videos, or walk in the actual environment. Using photos or videos is convenient for controlling variables, but lacks realism, and in a real environment, it is difficult to control variables because there are too many interfering factors. Experiment 2 of this study used VR technology to simulate a real environment, and in so doing, solved both problems; the 3D perspective is more realistic than 2D photos or videos, and it is easier to control variables in the simulated environment. The construction of a landscape with healing effects can not only relieve the pressure on an urban population and prevent anxiety, depression, and other mental diseases, but also provides a new treatment method for patients with anxiety and depression. This approach thus combines traditional medicine, psychological consultation, and natural intervention therapy, helping patients to improve their depression and anxiety in a more effective way.

### 4.3. Limitations and Opportunities for Future Research

Although this study has taken a small step in the design of patient-oriented health landscapes, there is still much to be improved in future empirical research on health landscapes.

In terms of experimental design, first of all, this experiment used transient experiments instead of long-term experiments. If long-term landscape intervention therapy can be administered to patients, a control group will be formed for the medication treatment group, and the anxiety and depression scales will be used to evaluate their depression, anxiety and stress levels. It would be more meaningful clinically. Secondly, the duration of each landscape experience needs to be extended to 3–6 min, which is not long enough to cause differential changes in physiological indicators. Finally, the independent variables of sight distance and sparse density in this study were preset categorical variables. It is suggested that continuous variables be used that allow people to walk freely in the environment and facilitate measurement of physiological indicators from time to time, so as to find the most comfortable and relaxing landscape types and identify various landscape indicators.

In terms of research subjects, this article was aimed at patients with depression. During the experimental investigation, it was found that the number of patients with anxiety disorder was much higher than that of patients with depression. Anxiety disorder patients are a more common and high-risk group in society, and anxiety disorders are more common to the outside world. The environment is more sensitive, and improvements in the environment may be more significant for mood improvement and the treatment of patients with anxiety disorder.

## 5. Conclusions

This research was an initial attempt to explore the impacts of urban park types, viewing distance, and edge permeability on depressive patients’ anxiety, depression, and stress. Study A used six types of landscape to examine the perceived anxiety, depression and stress before and after patient viewing of photos. The results confirmed our hypothesis: The grassland landscape has the optimal restorative effect on patients with depressive disorders. In study B, patients with depression were randomly assigned to be immersed in nine natural environments with different viewing distances and different permeability levels. The immersed virtual environment (IVE) was constructed with VR equipment. The physiological effects of participants were measured continuously through the whole experiment. The psychological effects on the participants were measured before and after the experiment and provided evidence-based guidelines for therapeutic landscape design. The results revealed that the grassland landscape with a higher viewing distance and medium edge permeability has a higher restorative effect on patients with depressive disorders.

Our results show that urban park landscapes play an important role in the restoration of mental health for patients with depressive disorders. Human society has been centered upon high-density and fast-paced urban environments for a long time, which has induced a number of diseases, especially mental diseases, such that health problems have become national and worldwide concerns. The United States and the United Kingdom have issued guidelines for healthy city design. The research trends in future landscape design and guidelines will gradually shift from aesthetics and ecology to health restoration. Finally, the most significant contribution of this study might be that it provides a method of nature therapy to improve the mental health problems of patients with depressive disorders. Nature prescription programs are increasing in popularity around the world [5]. Psychiatrists may be able to use nature prescriptions for depression treatment, combined with traditional medication.

## Figures and Tables

**Figure 1 ijerph-19-09867-f001:**
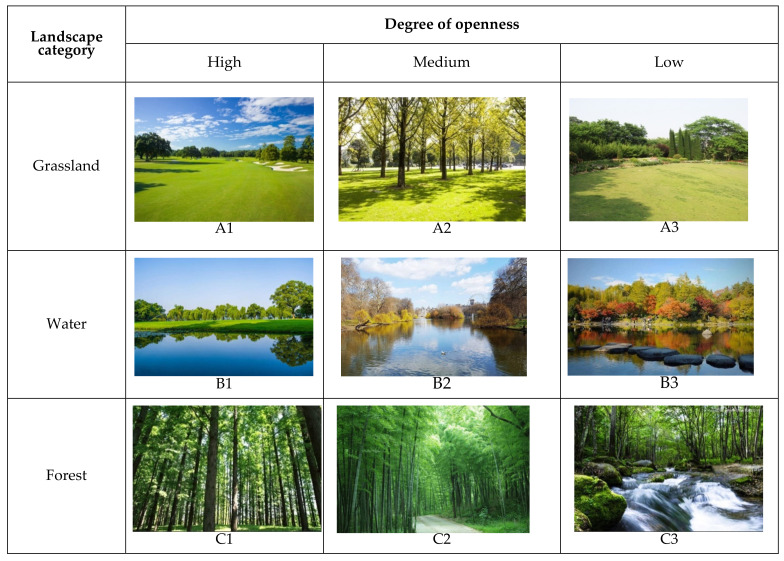
Representative photos for the six landscape categories.

**Figure 2 ijerph-19-09867-f002:**
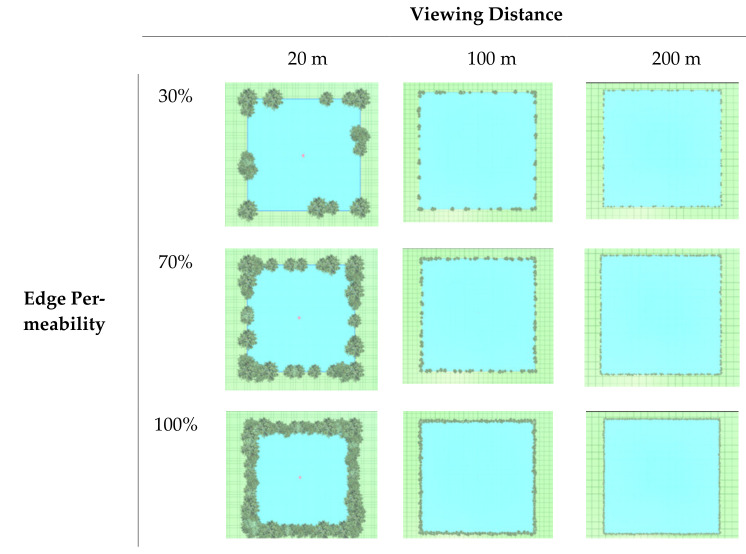
VR landscape scene design plans.

**Figure 3 ijerph-19-09867-f003:**
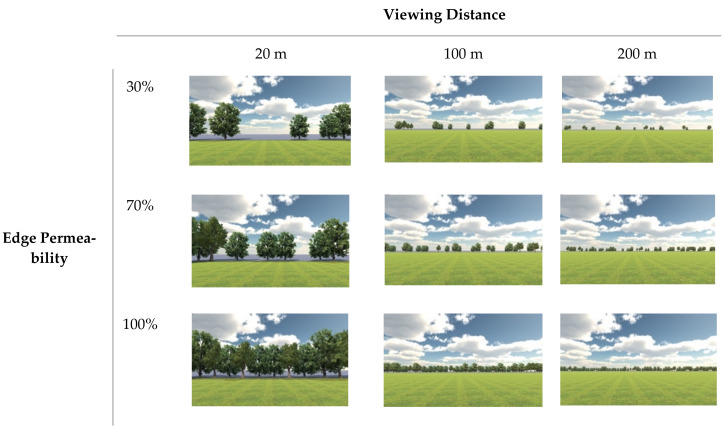
VR landscape scene renderings.

**Figure 4 ijerph-19-09867-f004:**
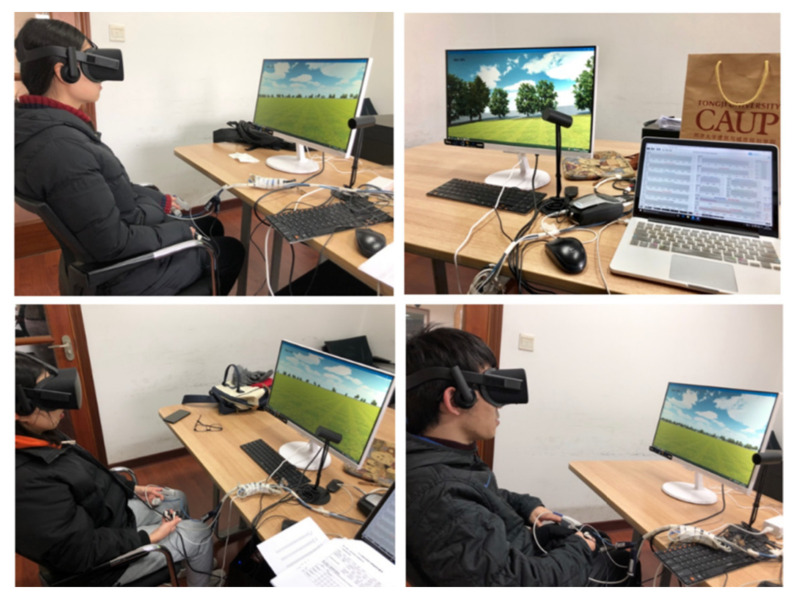
Participants were immersed in a virtual natural environment.

**Figure 5 ijerph-19-09867-f005:**
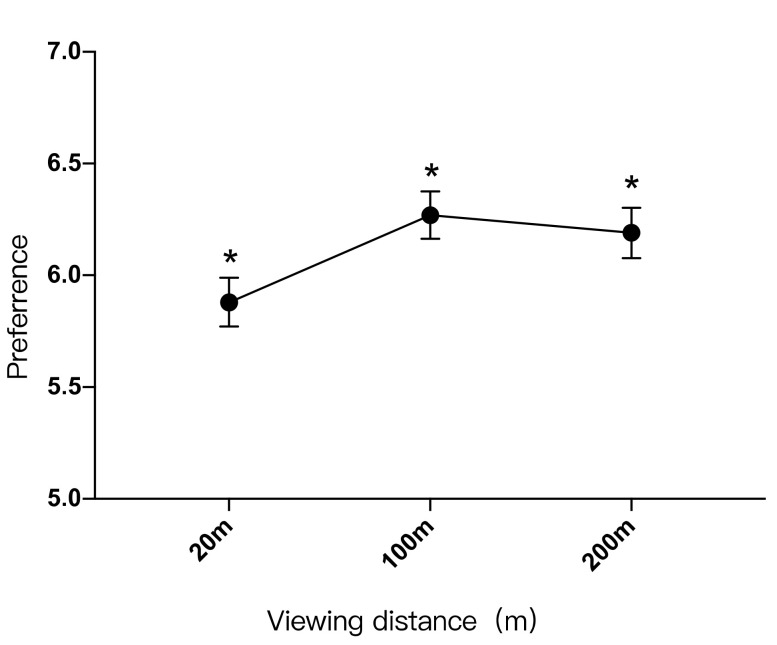
Impacts of different viewing distances on perceived preference, *: *p* < 0.05.

**Figure 6 ijerph-19-09867-f006:**
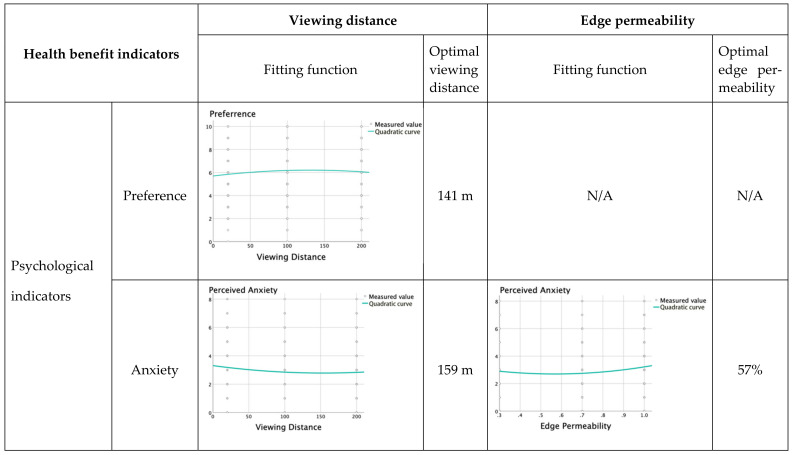
Optimal viewing distance and edge permeability of the grassland landscape.

**Figure 7 ijerph-19-09867-f007:**
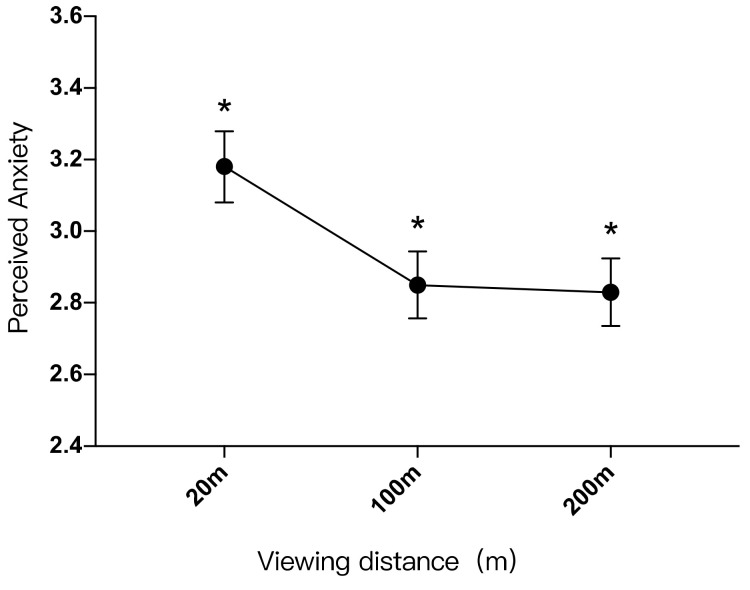
Impacts of different viewing distances on perceived anxiety, *: *p* < 0.05.

**Figure 8 ijerph-19-09867-f008:**
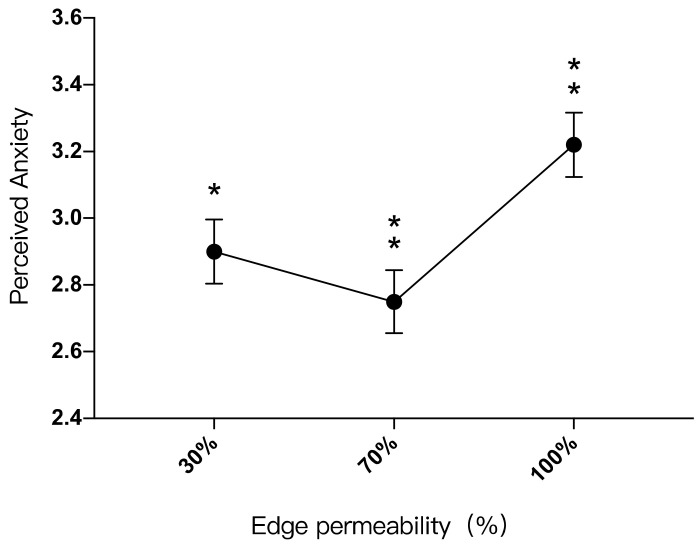
Impacts of different edge permeability levels on perceived anxiety, *: *p* < 0.05, **: *p* < 0.01.

**Figure 9 ijerph-19-09867-f009:**
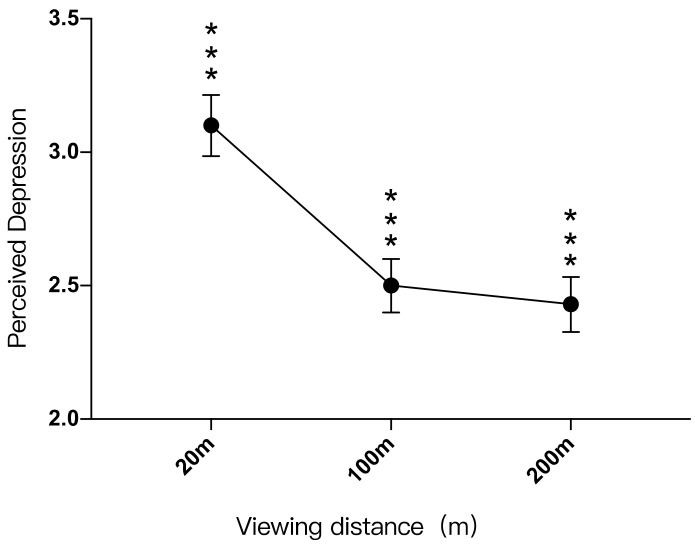
Impacts of different viewing distances on perceived depression, ***: *p* < 0.001.

**Figure 10 ijerph-19-09867-f010:**
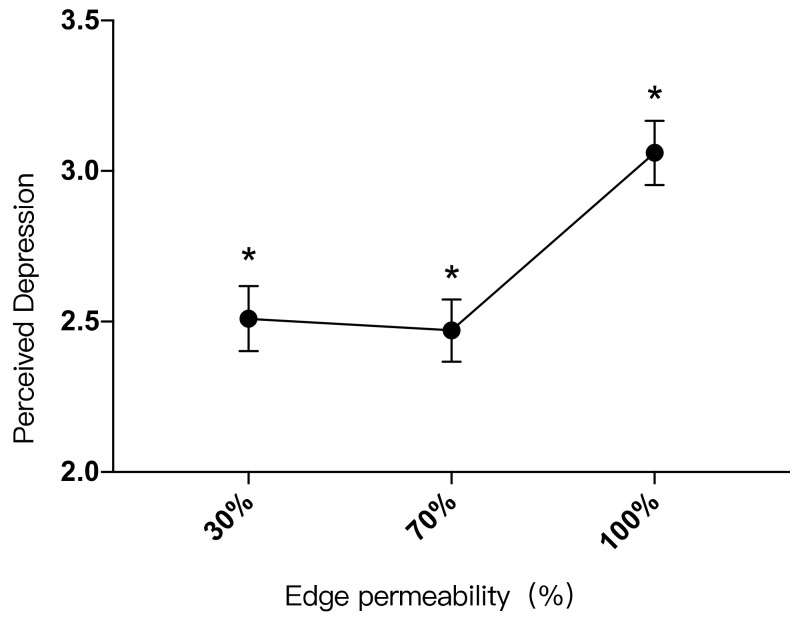
Impacts of different edge permeability levels on perceived depression, *: *p* < 0.05.

**Table 1 ijerph-19-09867-t001:** Procedure of study A. (The procedure includes two stages: The first stage is for preparation; the second stage is experiment conduction. The whole process needs 20 min).

Preparation (4 min)			Experiment (16 min)		
1 min	2 min	1 min	10 min	1 min	5 min
Sign informed consent	Introduce the objective and requirements of the experiment	Fill in basic personal information	Evaluate anxiety and depression	Contemplate landscape photos	Fill in the landscape preference scale

**Table 2 ijerph-19-09867-t002:** Procedure of study B. (The procedure includes three stages: The first stage is for preparation; the second stage is experiment conduction, the final stage is the end. The whole process needs 35 min).

Preparation Stage (10 min)				Experimental Stage (24 min)					End (1 min)
1 min	3 min	5 min	1 min	1 min	2 min	1 min	2 min	18 min	1 min
Fill in basic personal information	Introduce objective of the experiment	Connect instrument	Close eyes and relax	Pressure guidance	Fill in the psychological scale	Pressure relief (VR experience)	Fill in the psychological scale	Repeat experimental steps for total of 9 images	Receive the reward

**Table 3 ijerph-19-09867-t003:** Impacts of different viewing distances and permeability levels on △E_SC._

	df ^a^	MS ^b^	F	*p* ^c^	η ^ **2** ^ _ **p** _ ^ **d** ^
viewing distance	2	0.000	0.079	0.924	0.000
permeability	2	0.001	0.118	0.888	0.000
viewing distance × permeability	4	0.004	0.865	0.484	0.004

^a^ Degree of freedom, ^b^ Mean square, ^c^ *p* value, ^d^ effect size.

**Table 4 ijerph-19-09867-t004:** Impacts of different viewing distances and permeability levels on △E_ST._

	df ^a^	MS ^b^	F	*p* ^c^	η ^ **2** ^ _ **p** _ ^ **d** ^
viewing distance	2	0.000	1.538	0.215	0.003
permeability	2	0.000	0.983	0.375	0.002
viewing distance × permeability	4	0.001	0.260	0.903	0.001

^a^ Degree of freedom, ^b^ Mean square, ^c^ *p* value, ^d^ effect size.

**Table 5 ijerph-19-09867-t005:** Impacts of different viewing distances and permeability levels on △E_HR._

	df ^a^	MS ^b^	F	*p* ^c^	η ^ **2** ^ _ **p** _ ^ **d** ^
viewing distance	2	0.001	0.131	0.877	0.000
permeability	2	0.003	0.686	0.504	0.002
viewing distance × permeability	4	0.002	0.571	0.684	0.003

^a^ Degree of freedom, ^b^ Mean square, ^c^ *p* value, ^d^ effect size.

**Table 6 ijerph-19-09867-t006:** Impacts of different viewing distances and permeability levels on △E_BVP Amp._

	df ^a^	MS ^b^	F	*p* ^c^	η ^ **2** ^ _ **p** _ ^ **d** ^
viewing distance	2	0.108	0.912	0.971	0.002
permeability	2	0.004	0.030	0.402	0.000
viewing distance × permeability	4	0.070	0.588	0.671	0.003

^a^ Degree of freedom, ^b^ Mean square, ^c^ *p* value, ^d^ effect size.

**Table 7 ijerph-19-09867-t007:** Impacts of different viewing distances and permeability levels on △E_SPO2._

	df ^a^	MS ^b^	F	*p* ^c^	η ^ **2** ^ _ **p** _ ^ **d** ^
viewing distance	2	2.327 × 10^−6^	0.099	0.906	0.000
permeability	2	7.159 × 10^−6^	0.304	0.738	0.001
viewing distance × permeability	4	7.994 × 10^−6^	0.339	0.852	0.002

^a^ Degree of freedom, ^b^ Mean square, ^c^ *p* value, ^d^ effect size.

**Table 8 ijerph-19-09867-t008:** Impacts of different viewing distances and permeability levels on △E_RMSSD._

	df ^a^	MS ^b^	F	*p* ^c^	η ^ **2** ^ _ **p** _ ^ **d** ^
viewing distance	2	0.160	0.770	0.463	0.002
permeability	2	0.348	1.673	0.188	0.004
viewing distance × permeability	4	0.170	0.819	0.513	0.004

^a^ Degree of freedom, ^b^ Mean square, ^c^ *p* value, ^d^ effect size.

**Table 9 ijerph-19-09867-t009:** Impacts of different viewing distances and permeability levels on △E_SDNN._

	df ^a^	MS ^b^	F	*p* ^c^	η ^ **2** ^ _ **p** _ ^ **d** ^
viewing distance	2	0.130	1.398	0.248	0.003
permeability	2	0.204	0.891	0.411	0.002
viewing distance × permeability	4	0.209	1.439	0.219	0.007

^a^ Degree of freedom, ^b^ Mean square, ^c^ *p* value, ^d^ effect size.

**Table 10 ijerph-19-09867-t010:** Impacts of different viewing distances and permeability levels on △E_LF/HF._

	df ^a^	MS ^b^	F	*p* ^c^	η ^ **2** ^ _ **p** _ ^ **d** ^
viewing distance	2	0.033	0.022	0.978	0.000
permeability	2	0.136	0.089	0.915	0.000
viewing distance × permeability	4	4.704	2.431	0.046 *	0.011

^a^ Degree of freedom, ^b^ Mean square, ^c^ *p* value, ^d^ effect size, *: *p* < 0.05.

**Table 11 ijerph-19-09867-t011:** Summary of the impacts of viewing distance and edge permeability of a grassland landscape on psychological health.

Health Benefit Indicators		Viewing Distance	Edge Permeability	Viewing Distance × Edge Permeability
Physiological indicators	GSR	N	N	N
	HR	N	N	N
	BVP Amp	N	N	N
	SPO2	N	N	N
	RMSSD	N	N	N
	SDNN	N	N	N
	LF/HF	N	N	N
Psychological indicators	Preference	+	N	N
	Safety	N	N	N
	Anxiety	+	++	N
	Depression	+++	+++	N

N, no impact; “+”, slightly significant impact (*p* ≤ 0.05); “++”, significant impact (*p* ≤ 0.01); “+++”, extremely significant impact (*p* ≤ 0.001). The color represents different degree of significant impact, the level of significant impact increases with the color getting darker.

**Table 12 ijerph-19-09867-t012:** Impacts of different viewing distances and permeability levels on perceived preference.

	df ^a^	MS ^b^	F	*p* ^c^	η ^ **2** ^ _ **p** _ ^ **d** ^
viewing distance	2	12.927	3.734	0.024	0.009
permeability	2	6.600	1.906	0.149	0.004
viewing distance × permeability	4	1.106	0.320	0.865	0.001

^a^ Degree of freedom, ^b^ Mean square, ^c^ *p* value, ^d^ effect size.

**Table 13 ijerph-19-09867-t013:** Post hoc (LSD) multiple factors comparison analysis of impacts of different viewing distances on perceived preference.

Group (I vs. J)	Mean difference (I–J)	*p*
20 m vs. 100 m	−0.396 *	0.010
20 m vs. 200 m	−0.315 *	0.041
100 m vs. 200 m	0.081	0.600

*: *p* < 0.05.

**Table 14 ijerph-19-09867-t014:** Impacts of different viewing distances and permeability on perceived anxiety.

	df ^a^	MS ^b^	F	*p* ^c^	η ^ **2** ^ _ **p** _ ^ **d** ^
viewing distance	2	11.584	4.266	0.014	0.010
permeability	2	16.937	6.237	0.002	0.014
viewing distance × permeability	4	1.276	0.470	0.758	0.002

^a^ Degree of freedom, ^b^ Mean square, ^c^ *p* value, ^d^ effect size.

**Table 15 ijerph-19-09867-t015:** Post hoc (LSD) multiple factors comparison analysis of impacts of different viewing distances on perceived anxiety.

Group (I vs. J)	Mean Difference (I–J)	*p*
20 m vs. 100 m	0.333 *	0.014
20 m vs. 200 m	0.350 *	0.010
100 m vs. 200 m	0.017	0.901

*: *p* < 0.05.

**Table 16 ijerph-19-09867-t016:** Post hoc (LSD) multiple factors comparison analysis of impacts of different edge permeability levels on perceived anxiety.

Group (I vs. J)	Mean Difference (I–J)	*p*
30% vs. 100%	−0.316 *	0.020
70% vs. 100%	−0.468 **	0.001
30% vs. 70%	0.152	0.264

*: *p* < 0.05, **: *p* < 0.01.

**Table 17 ijerph-19-09867-t017:** Impacts of different viewing distances and edge permeability levels on perceived depression.

	df ^a^	MS ^b^	F	*p* ^c^	η ^ **2** ^ _ **p** _ ^ **d** ^
viewing distance	2	40.924	12.498	0.000	0.028
permeability	2	32.728	9.995	0.000	0.022
viewing distance × permeability	4	1.282	0.392	0.815	0.002

^a^ Degree of freedom, ^b^ Mean square, ^c^ *p* value, ^d^ effect size.

**Table 18 ijerph-19-09867-t018:** Post hoc (LSD) multiple factors comparison analysis of impacts of different viewing distances on perceived depression.

Group (I vs. J)	Mean Difference (I–J)	*p*
20 m vs. 100 m	0.603 ***	0.000
20 m vs. 200 m	0.677 ***	0.000
100 m vs. 200 m	0.074	0.621

***: *p* < 0.001.

**Table 19 ijerph-19-09867-t019:** Post-hoc (LSD) multiple factors comparison analysis of impacts of different edge permeability levels on perceived depression.

Group (I vs. J)	Mean Difference (I–J)	*p*
30% vs. 100%	−0.556 *	0.000
70% vs. 100%	−0.593 *	0.000
30% vs. 70%	0.037	0.805

*: *p* < 0.05.

**Table 20 ijerph-19-09867-t020:** Guidelines for healthy landscape design for patients with general depressive disorder.

Users	Healthy Design Indicators	Guidelines
Patients with depressive disorder	Landscape type	Grassland Landscape > Forest Landscape > Water Landscape > Forest Landscape > City Square Landscape Try to create a sense of wilderness and original ecological landscape
	Openness	Avoid closed spaces
	Viewing distance	between 141–159 m
	Edge permeability	between 52–57%
	Elements	Grassland > Water >Arbor > Tree Array Blue sky, sunshine, rich flora and fauna, sheltered space
	Color	Green > blue > purple > yellow

## Data Availability

Not applicable.

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
