# Peer review of "Impacts of Landscape Type, Viewing Distance, and Permeability on Anxiety, Depression, and Stress"

_ijerph, 2022, doi:10.3390/ijerph19169867_

Round 1

Reviewer 1 Report

This research conducted two experiments to investigate the effects of different landscapes on patients suffering from depression. The research topic is relevant and the study is interesting. However, I feel that the paper in its current format needs improvement. My main concerns are as follows:

1. The introduction does not provide sufficient background. In my opinion, authors could include more references to enlighten readers about the foundation of the research questions (page 2, lines 47-56). In addition, most of the literature cited by the author is from about ten years ago. Is there any recent related research? Meanwhile, the article lacks citations in many places. For example, in page 2, lines 59-61, the authors said that “a large number of studies have shown that the natural environment can effectively alleviate anxiety, help healthy people to recover from pressure, and promote cognitive ability and attention.” Authors need to cite relevant articles here.

2. I propose that the authors provide the results of Study 1 immediately following the method of Study 1, rather than presenting the method of Study 2. This will make the article easier to read, because the relationships between Studies 1 and 2 are gradual.

3. For methods, in page 5, lines 165, 1-2 patients participating in the pilot study seem to be too few. In addition, the authors should report the exact number of people participating in the pre-experiment, not a range “1-2”.

4. The reporting of results lacked core data and statistical analysis. For examples, in page 7, lines 241-242, “There was almost no difference between patients 241 with different degrees of depression”, authors should provide the statistical results of different groups and report the related statistical analysis results; in page 9, lines 284-321, the authors seem to have done regression analyses, so you should report not only the regression function but also degree of freedom, P value, effect size, etc.

I think that the current presentation of all the results lacks scientific credibility. Maybe a researcher with a statistical background (e.g., mathematician, psychologist, and sociologist) can help you better analyze the data and present the results. In sum, I hope the authors rework the Results sections (both Study 1 and Study 2) with more detailed data and statistical analysis rather than simply publishing the conclusions you summarized.

Author Response

Point 1: The introduction does not provide sufficient background. In my opinion, authors could include more references to enlighten readers about the foundation of the research questions (page 2, lines 47-56). In addition, most of the literature cited by the author is from about ten years ago. Is there any recent related research? Meanwhile, the article lacks citations in many places. For example, in page 2, lines 59-61, the authors said that “a large number of studies have shown that the natural environment can effectively alleviate anxiety, help healthy people to recover from pressure, and promote cognitive ability and attention.” Authors need to cite relevant articles here.

Response 1: We’ve added more references to support our argument as the foundation of the research questions (Line 25-129).

Point 2: I propose that the authors provide the results of Study 1 immediately following the method of Study 1, rather than presenting the method of Study 2. This will make the article easier to read, because the relationships between Studies 1 and 2 are gradual.

Response 2: We’ve provided the results of Study 1 immediately following the method of Study 1.

Point 3: For methods, in page 5, lines 165, 1-2 patients participating in the pilot study seem to be too few. In addition, the authors should report the exact number of people participating in the pre-experiment, not a range “1-2”.

Response 3: Sorry, we made a typo. The pilot study includes 12 participants.

Point 4: The reporting of results lacked core data and statistical analysis. For examples, in page 7, lines 241-242, “There was almost no difference between patients 241 with different degrees of depression”, authors should provide the statistical results of different groups and report the related statistical analysis results;

Response 4: We didn’t do ANOVA analysis in study A. The meaning we want to express is that for patients with different degrees of depression, the results are the same.(Line 281-282)

Reviewer 2 Report

- The background needs to address how urbanization impacts chronic pressure. 

- Literature review needs to be added.

- Why is it important to study the virtual nature type, viewing distance, and edge permeability impacts on patients with depression and anxiety disorders?

- I do not think that exposure to images is sufficient for this type of study. Physical presence involves all six senses while being exposed to a photo of the landscape only relies on the visual sense only.

- The evaluation of anxiety and depression was totally subjective in study 1, while it could be informative to pair it with physiological measures.

- How was the number of participants determined for the study? What is the sampling method?

- I think it would be best to compare a physical space and a virtual space.

- The literature review needs to address similar studies and the discussion would reveal how this study is different or similar to findings from previous studies.

- The diagrams and figures are helpful to understand the study. 

- Why does the study have two experiments. Maybe comparing the results of both experiments would be helpful.

Author Response

Point 1: The background needs to address how urbanization impacts chronic pressure.

Response 1: We’ve rearranged the background part. (Line 25-129)

Point 2: Literature review needs to be added.

Response 2: Literature review has been added.

Point 3: Why is it important to study the virtual nature type, viewing distance, and edge permeability impacts on patients with depression and anxiety disorders?

Response 3: The reason has been presented from line 119-129.

Point 4: I do not think that exposure to images is sufficient for this type of study. Physical presence involves all six senses while being exposed to a photo of the landscape only relies on the visual sense only.

Response 4: The supporting evidence has been added from line 114-118.

Point 5: The evaluation of anxiety and depression was totally subjective in study 1, while it could be informative to pair it with physiological measures.

Response 5: Yes, study 1 is a preliminary experiment so we only use the perveived scale to measure the psycological health.

Point 6: How was the number of participants determined for the study? What is the sampling method?

Response 6: We’ve added the sampling method from line 321-325.

Point 7: I think it would be best to compare a physical space and a virtual space.

Response 7: Yes, you’re right, but this is not the objective of our research.

Point 8: The literature review needs to address similar studies and the discussion would reveal how this study is different or similar to findings from previous studies.

Response 8: We’ve added the difference from line 119-129.

Point 9: The diagrams and figures are helpful to understand the study.

Response 9: Thank you for your praise.

Point 10: Why does the study have two experiments. Maybe comparing the results of both experiments would be helpful.

Response 10: Study A is a preliminary experiment in order to find out which type of the urban park landscape has the optimal restoration level on anxiety, stress, and depression. Study B is based on study A in order to expore which range of viewing distance and edge permeability of this type of  landscape has the optimal restoration level on anxiety, stress, and depression.

Reviewer 3 Report

Dear Authors, I have reviewed the manuscript "Impacts of landscape type, viewing distance, and permeability on anxiety, depression, and stress", which aimed to study what kind of urban park landscapes can alleviate depression, anxiety, and stress on patients with depression, and to provide suggestions on therapeutic landscape design. . Prior to further processing, the following observations and comments should be addressed:

- The summary needs to be rewritten. The sentences do not have a logical connection. Before the objective you could include a sentence about why they set that objective. 

- In the introduction you should generate at least 5 paragraphs. Remember the paragraphs 

- Do not separate the introduction by subheadings. Only generate paragraphs about those subheadings in a narrative way and with a logical connection.

- Include an opening paragraph that talks about the landscape and its importance focused on the objectives of your study.

- Eliminate bullet points in the last paragraph of the introduction. Generate a paragraph in narrative form that describes the problem, the questions that are generated from this and the objectives that are set to respond.

- The first part of the methodology should be transferred to the introduction, final paragraph.

- Initially, the methodology should describe the design, approach and techniques used. Next, describe the procedure considering the objectives of your study.

- The methodology should be synthesized.

When you define the objectives clearly, you will be able to structure the results precisely.

- The design of the study should be significantly improved, in the current state it is difficult to review.

- Try not to use too many vignettes.

- Conclusions again appear objective. The conclusion is should respond to the objectives stated in the introduction and describe the limitations, together with future research from your results.

Author Response

Point 1: The summary needs to be rewritten. The sentences do not have a logical connection. Before the objective you could include a sentence about why they set that objective.

Response 1: We have added the sentence about why we set that objective (page 1, line 10-14).

Point 2: In the introduction you should generate at least 5 paragraphs. Remember the paragraphs. Do not separate the introduction by subheadings. Only generate paragraphs about those subheadings in a narrative way and with a logical connection.

Include an opening paragraph that talks about the landscape and its importance focused on the objectives of your study.

Eliminate bullet points in the last paragraph of the introduction. Generate a paragraph in narrative form that describes the problem, the questions that are generated from this and the objectives that are set to respond.

The first part of the methodology should be transferred to the introduction, final paragraph.

Response 2: We have deleted the subheadings, added opening paragraph (Line 25-34). Bullet points have been eliminated. We’ve generated paragraphs in a narrative way and with a logical connection. (Line 25-140).Because we two studies, if we transfer the first of methodology into inrtoduction, it will be not clear, so we prefer to keep this part in methodology.

Point 3: Initially, the methodology should describe the design, approach and techniques used. Next, describe the procedure considering the objectives of your study. The methodology should be synthesized. When you define the objectives clearly, you will be able to structure the results precisely. The design of the study should be significantly improved, in the current state it is difficult to review. Try not to use too many vignettes.

Response 3: We have already described the design, approach, techniques and objectives in methodology.

Point 4: Conclusions again appear objective. The conclusion is should respond to the objectives stated in the introduction and describe the limitations, together with future research from your results.

Response 4: We’ve added the objective and results into conclusion part. The limitations and future research is described in the discussion part.

Round 2

Reviewer 1 Report

The issues in the Introduction and Results sections have been greatly improved by the authors. After addressing some minor issues, I believe this paper is suitable for publication.

1.     A direct comparison of percentages is not appropriate in the results section of Study 1. Using the chi-square test here can help to provide more scientific results.

2.     Abbreviations are used inappropriately in the manuscript. First, abbreviations that have not appeared before should not be used directly in the text. For example, page 9 line 319, “TSST” should be replaced with “trier social stress test (TSST)”. Second, there is no need to provide abbreviated forms for terms mentioned only once in the manuscript, e.g., “skin electricity (SC)”, “skin temperature (ST)”. Authors are expected to proofread the entire document and standardize the use of abbreviations.

Author Response

The issues in the Introduction and Results sections have been greatly improved by the authors. After addressing some minor issues, I believe this paper is suitable for publication.

Point 1: A direct comparison of percentages is not appropriate in the results section of Study 1. Using the chi-square test here can help to provide more scientific results.

Response 1: Thank you for your advise. We’ve used the chi-square test to do data analysis in Study A to justify “There was almost no difference between patients 241 with different degrees of depression”. (Line 292-294)

Point 2: Abbreviations are used inappropriately in the manuscript. First, abbreviations that have not appeared before should not be used directly in the text. For example, page 9 line 319, “TSST” should be replaced with “trier social stress test (TSST)”. Second, there is no need to provide abbreviated forms for terms mentioned only once in the manuscript, e.g., “skin electricity (SC)”, “skin temperature (ST)”. Authors are expected to proofread the entire document and standardize the use of abbreviations.

Response 2: Thank you for your reminder. We’ve corrected the Abbreviations. For “skin electricity (SC)” and “skin temperature (ST)”, we not only used once because we also used the ΔESC and ΔEST in the results part. So we prefer to keep the Abbreviations here.

Reviewer 2 Report

The introduction and conclusion can be refined.

Author Response

Point 1: The introduction and conclusion can be refined.

Response 1: Thank you for your advise. We’ve refined the introduction and conclusion part. (Line 34-37,1047-1053)

Reviewer 3 Report

I have reviewed the new version and the authors have complied with the requested changes.

Author Response

Thank you for your approval!